# Phacoemulsification Induced Changes of Choroidal Thickness in Eyes with Age-Related Macular Degeneration

**DOI:** 10.3390/medicina56050252

**Published:** 2020-05-22

**Authors:** Gailė Gudauskienė, Ieva Povilaitytė, Eglė Šepetauskienė, Dalia Žaliūnienė

**Affiliations:** 1Department of Ophthalmology, Medical Academy, Lithuanian University of Health Sciences, LT-50009 Kaunas, Lithuania; ieva.povilaityte@fc.lsmuni.lt (I.P.); dalia.zaliuniene@lsmu.lt (D.Ž.); 2Information Technology Centre, Lithuanian University of Health Sciences, LT-50009 Kaunas, Lithuania; egle.sepetauskiene@lsmuni.lt

**Keywords:** phacoemulsification, choroidal thickness, age-related macular degeneration

## Abstract

*Background and Objectives*: Patients with cataract and age-related macular degeneration (AMD) may safely undergo cataract phacoemulsification to enhance visual acuity. Although it has not been proven that cataract surgery can cause AMD progression, different phacoemulsification effects are observed not only on retinal but also on choroidal tissues. The purpose of this study was to evaluate the effect of phacoemulsification on the choroidal thickness (CT) in eyes with and without AMD. *Materials and Methods*: In 32 eyes of 32 patients with senile cataract (No-AMD group) and in 32 eyes of 32 patients with cataract and dry AMD (AMD group), who had phacoemulsification without intraoperative complications and intraocular lens implantation, foveal retinal thickness (FRT) and CT were evaluated three times: at 1–2 post meridiem preoperatively, then 1 month and 3 months postoperatively, using 1050 nm swept source-optical coherence tomography (Topcon, Tokyo, Japan). *Results*: In both groups, a significant increase in FRT was observed after one month and a decrease after three months without reaching the baseline. One month after surgery, a sectorial CT increase was apparent in all sectors in both groups. A negative association between CT and age was disclosed in the No-AMD group almost for all regions at all time points. Furthermore, CT was significantly negatively associated with axial length (AL) in all sectors at all time points in the AMD group. *Conclusion*: Uneventful phacoemulsification may induce changes in the posterior eye segment. An increase in CT and FRT was observed in both groups one month after the surgery. However, three months after surgery, CT changes were different in both groups, while FRT decreased in both groups. CT changes negatively associated with age in the No-AMD group and with AL in the AMD eyes. These postoperative changes in the choroid and retina may not only lead to the late-onset pseudophakic cystoid macular edema but also to progression of AMD.

## 1. Introduction

Phacoemulsification is a modern extracapsular cataract extraction involving emulsification of the lens with an ultrasonic energy and aspiration from the eye. However, phacoemulsification provokes discharge of pro-inflammatory molecules not only affecting the anterior but also posterior eye segments [1]. In some cases, the inflammation can lead to corneal edema and pseudophakic cystoid macular edema (PCME).

Choroidal structure has been evaluated in clinical investigations with controversial results [2]. Some authors have confirmed an increase in the choroidal thickness (CT) after cataract phacoemulsification [3,4]. Others, however, estimated opposite outcomes [5,6]. Literature data is shown in Table 1.

Cataract is a major reason of world blindness and visual impairment caused by the clouding of the natural eye lens [7]. The numbers of cataract blindness worldwide raised from 12.3 to 20 million during 20 years until 2010 [8].

The other sight-threatening disease—age-related macular degeneration (AMD) is a progressive degenerative pathology resulting in gradual worsening of eyesight, central vision loss and complete blindness. The dry form accounts for 90% of AMD events [9]. It is diagnosed by the presence of drusen and the damage of the retinal pigment epithelial (RPE) cells. Although early AMD stages are frequently asymptomatic, later on patients often notice missing areas of vision, wavy lines, central scotoma, and continuous deterioration of visual acuity [10]. It is disagreed whether the choroid is primarily affected in concert with the retina or secondarily involved in the pathological mechanism [11].

The choroid supplies nutrients and oxygen to the outer retinal layers, regulates ocular metabolism and temperature as a vascular eye structure. The macula is crucial for the central and color vision and is dependent on choroidal blood flow [12]. Intraocular pressure (IOP) fluctuations tend to change macular volume and thickness and may cause micro ischemia in retinal nerve fiber layer and micro vessels [8]. It has been published that a moderate decrease in IOP can be observed after phacoemulsification and may influence the ocular perfusion pressure (OPP) [9]. Furthermore, the calculated OPP may be used to evaluate the eye vascular changes [10].

Swept source-optical coherence tomography (SS-OCT) uses a 1050-nm laser beam as a light source and has high imaging speed [7,13]. This novel high-speed OCT system utilizes laser of longer wavelength range, which penetrates deeper into the choroidal tissue [14]. It ensures better visualization and evaluation of choroidal changes [15,16]. SS-OCT offers improvements in visualizing the vitreous, retina, choroid and sclera. The increased scan speeds, decreased signal attenuation and deeper tissue penetration make SS-OCT ideal for capturing wide fields of view and for studying structures below the RPE, especially the choroid [17].

The purpose of this study was to evaluate the effect of phacoemulsification without intraoperative complications on the choroidal thickness in eyes with and without dry AMD. To the best of our knowledge, choroidal thickness after cataract surgery has been studied in patients with dry AMD for the first time.

## 2. Materials and Methods

We conducted a prospective clinical study of 32 patients (32 eyes) with dry AMD (AMD group) and 32 patients (32 eyes) without AMD (No-AMD group), who had phacoemulsification surgery without intraoperative complications for senile cataract performed at the Department of Ophthalmology of the Hospital of Lithuanian University of Health Sciences, Kaunas Clinics. All patients were informed and signed a consent form. The study was approved by the Kaunas Regional Biomedical Research Ethics Committee 2015-07-09 No. BE-2-26 and 2017-01-26 No. P1-BE-2-26/2015 and matched requirements of the Declaration of Helsinki.

The inclusion criterion was a visually significant cataract with or without dry AMD. AMD was classified according to the Age-Related Eye Disease Study [18]. Early mild AMD lesions were defined by plural tiny drusen and several intermediate drusen (63–124 μm in diameter) or RPE changes. Early intermediate AMD was described by broad intermediate drusen and at least one large druse (≥125 μm diameter) or geographic atrophy (GA) not reaching the central fovea, while healthy controls were described by less than five or no drusen. Advanced AMD eyes characterized by geographic atrophy involving the fovea and/or any of the features of neovascular AMD were not included in the study.

Patients with severe cataracts that could cause poor SS-OCT quality (an image quality index <40, motion artifacts, segmental loss in image signal, indistinct vessel boundaries) and patients with refractive errors greater than 3 diopters, unstable fixation, glaucoma, other ocular abnormalities, traumatic cataract, diabetes mellitus, any ocular surgery, corticosteroid therapy, increased postoperative IOP (>21 mmHg) were excluded.

A full ocular examination was performed by an experienced ophthalmologist before surgery, including best corrected visual acuity (BCVA) test using LogMAR chart, measurement of IOP with Goldmann applanation tonometry, biomicroscopy, fundus examination, axial length (AL) measurement using Aladdin optical biometer based on optical low-coherence interferometry (Topcon, Tokyo, Japan). Cataract hardness was evaluated and graded based on the Lens Opacities Classification System III [19]. Optical biometry was performed for IOL power calculation (IOLMaster; Carl Zeiss Meditec Inc., Jena, Germany). All participants were asked about their medical history. The mean arterial pressure (MAP) was calculated using the systolic (SBP) and diastolic blood pressure (DBP) = DBP + 1/3 (SBP—DBP). The ocular perfusion pressure (OPP) was estimated = 2/3 MAP—IOP.

The cataract phacoemulsification with a clear cornea 2.4 mm incision was performed between September 2018 and July 2019 using a standard technique, under intravenous anesthesia. Phacoemulsification was performed by the same experienced surgeon using the Centurion^®^ Vision System (Alcon Laboratories, Inc., Geneve, Switzerland), the mean phaco energy was between 20% and 35%. An acrylic intraocular lens was implanted in the capsular bag in all operated eyes, no intraoperative complications were reported. All patients were treated with the same postoperative medication, consisting of dexamethasone eye drops (tapered weekly from 4 to 1 time per day), and levofloxacin eye drops (4 times daily for two weeks) postoperatively.

The OCT images were obtained after pupil dilation with tropicamide 1% using SS-OCT (DRI OCT Triton; Topcon, Tokyo, Japan) at 1050 nm wavelength between 1 and 2 post meridiem before surgery (P), one (M1) and three months (M3) after cataract surgery by the same experienced technician. A retinal–choroidal map was produced according to the Early Treatment Diabetic Retinopathy Study (ETDRS) grid nine subfields using automated software: the choroidal thickness (CT) and foveal retinal thickness (FRT) in the central ring (C), nasal inner (iNAS), superior inner (iSUP), temporal inner (iTEM), inferior inner (iINF), nasal outer (oNAS), superior outer (oSUP), temporal outer (oTEM) and inferior outer (oINF) regions. These scans were marked as the patient’s baseline, with “follow-up” function used in precisely the same locations.

Statistical analysis was performed with SPSS 23 program. The results were expressed as mean (standard deviation), (M (SD)). The Kolmogorov–Smirnov test was used for determination of quantitative data distribution. The Mann–Whitney U test was used for the scores of two independent groups. Differences between dependent variables (CT, FRT, IOP, OPP, BCVA changes) were analyzed using Wilcoxon Signed-Rank test and Friedman two-way analysis. Fisher‘s exact test was used to compare the frequencies of qualitative variables. Comparisons of CT in relation to age, sex, AL, preoperative BCVA and IOP were done by a simple linear regression test. Differences were considered statistically significant when *p* values ≤ 0.05.

## 3. Results

32 patients (32 eyes) with cataract and dry AMD and 32 patients (32 eyes) with cataract without AMD were involved in the study. The subjects’ distribution by sex, age, AL and PCME according to the occurrence of AMD is shown in Table 2.

Before surgery, BCVA did not differ between the groups, while M1 and M3 BCVA was better in the No-AMD group without retinal pathology (*p* = 0.129, *p* = 0.022 and *p* = 0.031, respectively, Mann–Whitney U test). There was a statistically significant improvement in BCVA after surgery in both groups (*p* < 0.05).

The postoperative IOP diminished throughout the follow-up period compared to preoperative values in both groups (*p* < 0.05). The mean P, M1 and M3 IOP between the groups did not differ (*p* = 0.366, *p* = 0.095 and *p* = 0.649, respectively, Mann–Whitney U test).

The mean OPP was higher at all time points in the No-AMD group (*p* = 0.006, *p* = 0.008 and *p* = 0.016, respectively, Mann–Whitney U test). Although there was a tendency for decrease in postoperative OPP values, OPP measurements at different follow-up points differed significantly in neither group (*p* > 0.05).

The FRT did not differ between the groups at any of the time points (*p* = 0.844, *p* = 0.768 and *p* = 0.906, respectively). There was a significant increase in FRT after 1 month; however, after 3 months a decrease without reaching the baseline was noticed in AMD patients. Table 3 lists changes in these parameters in the groups.

There were no statistically significant differences comparing CT in separate sectors during the different visits between the groups, although there was a tendency for thinner CT in AMD eyes (*p* > 0.05 by Mann–Whitney U test). One month after the surgery, the increase was apparent in all sectors in both groups. However, the significance was diverse. CT in the AMD group increased in all regions at both time points, but the significant difference was estimated in the outer nasal, inner superior, outer inferior regions one month after phacoemulsification (*p* = 0.043, *p* = 0.042, *p* = 0.01, respectively), and in the outer temporal, inner superior and both inferior regions three months after phacoemulsification (*p* = 0.017, *p* = 0.045, *p* = 0.035, *p* = 0.041, respectively). In No-AMD patients, an increase in CT was observed in the inner inferior and outer inferior regions (*p* = 0.023, *p* = 0.05, respectively), while 3 months postoperatively, the increase was apparent in the outer nasal, outer temporal, outer superior, inner inferior and outer inferior regions (*p* = 0.029, *p* = 0.039, *p* = 0.013, *p* = 0.034, *p* = 0.008, respectively) (Figure 1).

A single regression analysis of age, sex, AL, preoperative BCVA and IOP at three time points in both groups was performed. Univariate linear regression analysis disclosed negative significant associations between CT and AL (in all regions) and age (in several regions) at the studied time points in AMD patients. Furthermore, CT was significantly negatively associated with age in almost all sectors at the studied time points and several significant negative associations were observed with AL in No-AMD group. There were no associations between CT and sex, preoperative BCVA and IOP in neither group. Full-scale results are available in the online supplement (Appendix A).

## 4. Discussion

Cataract phacoemulsification is the most common ophthalmological surgery that may induce an inflammatory reaction in various eye tissues. In our current study, we found that following cataract surgery, not only retinal but also choroidal changes were induced. In our study, we assessed 32 cataract eyes with dry AMD and 32 eyes without AMD and determined that in the AMD group preoperative FRT was 248.64 (34.34), 1 month after surgery 269.72 (66.37), 3 months after surgery 252.36 (34.70) µm (*p* = 0.031), while in the No-AMD group it was 238.13 (24.54), 1 month after surgery 253.03 (42.81), 3 months after surgery 249.10 (30.42) µm (*p* = 0.001). Noda and associates have also reported a transient, albeit insignificant, postoperative increase in FRT [20]. Meanwhile, others have demonstrated a statistically significant increase after 1 and 3 months postoperatively [5,21,22]. This is similar to our results, as a significant increase after 1 month and a decrease in FRT without reaching the baseline after 3 months was observed.

In our study, although all surgeries were without intraoperative complications, a late postoperative complication PCME was observed in 3.1% patients in both groups one month after surgery, similarly to the data of Pierru et al. (2.6% patients) [23]. For all patients, no ocular and systemic risk factors for PCME were found. The inflammatory reaction, that could be one of the major reasons of the retinal thickness changes after phacoemulsification, may also lead to PCME formation.

Although in the AMD group there was a tendency for thinner CT in all sectors preoperatively, and almost in all sectors postoperatively, the differences were insignificant. This could be induced by loss of small choroidal vessels, narrowed choriocapillaris lumen, loss of cellularity and decreased blood flow. Choroidal structure investigations in AMD eyes may provide some of the clues in pathogenesis as abnormal choroidal circulation can contribute to the process of AMD. In other studies, the CT has been reported to be thinner in advanced AMD eyes [24] and especially closely related to the geographic atrophy progression among advanced AMD patients [25]. However, other authors have not found associations between non-exudative AMD form and alterations of the central choroid [26,27].

To the best of our knowledge, CT after cataract surgery has been studied in patients with dry AMD for the first time. In our research, we compared CT changes after cataract phacoemulsification in two different groups. We found a tendency for increased CT in all sectors after one month in both groups, and in several sectors results reached statistical significance. At three months after surgery, the increase was still observed in the AMD group, while the changes in No-AMD eyes were scattered. It is interesting that postoperative increase in CT was the greatest in the nasal and inferior regions at the second time point in the AMD group, because the baseline choroid was the thinnest in these regions. This relative thinning may be caused by isolated choroidal circulation and the closure of fetal fissure inferiorly leading these regions to vulnerability of IOP changes [28].

Other authors have analyzed postoperative CT changes in eyes without retinal pathology and found a significant increase in CT. Some authors have declared a prolonged significant increase in subfoveal CT (SFCT) after phacoemulsification [20,29,30]. Similar to our results, Abdellatif and coworkers have published that SFCT increased statistically significantly after 1 month (*p* = 0.014) with no significant change 3 months after the surgery (*p* = 0.073) [21]. As opposed to our results, other studies have announced no significant CT changes [5,6]. However, a different OCT device has been used or manual measurements analyzed in those studies. In the present study, we used an automated ETDRS grid that provides more detailed data for follow up choroidal map topography than single-point assessments.

Our study discovered a negative association between CT and age in the No-AMD group in almost all regions at all time points. In addition, CT was significantly negatively associated with AL in all sectors during each period in the AMD group. Similar to our No-AMD group data, Ozdogan et al. [31] published data showing a negative correlation between age and SFCT in healthy subjects. Other studies have also analyzed the relation between CT and various factors. Ohsugi et al. [32] determined a decrease in IOP after surgery, confirming the importance of IOP alterations for CT changes in the short term and published similar results to ours: no significant associations of CT were detected with preoperative IOP and BCVA. Jiang et al. [33] found a correlation between CT and sex, AL, IOP, and ultrasound (US) time. We did not record the US time in our study. On the other hand, Celik et al. [22] published different results. This confirms that the AL and IOP changes are crucial for the estimation of CT variation. As SFCT negatively correlates with myopic refractive error, a longer AL leads to a thinner choroid.

The CT increase may be influenced not only by inflammation, but also by enhanced light absorption and changes in IOP and OPP. There was a statistically significant decrease in IOP values after surgery in both groups. A decrease in IOP after cataract surgery has been reported in various studies [34,35,36].

There was a tendency for decreased postoperative OPP values; however, the OPP measurements at different postoperative follow up points differed significantly in neither group. Literature data shows opposite results of an increase in OPP due to the decrease in IOP [29]. In our study, there was a reduction in OPP after surgery in both groups, indicating not only a postoperative IOP change but also the MAP impact on OPP. Therefore, our results being different make us discuss other mechanisms of the CT increase, not only effects of the increased OPP.

Although local hormone eye drops were prescribed to all patients for one month postoperatively, after one month an increase in FRT and CT in all sectors was observed in all studied patients. This anti-inflammatory treatment could diminish the postoperative inflammation in the whole posterior segment, affecting pro-inflammatory mediators released during the surgery. Other authors have confirmed the lowering hormone effect on postoperative immune response and also on the rate of PCME [37]. However, one of the mediators (prostaglandins) may not only cause local inflammation but also increase uveoscleral outflow. This mechanism should be also considered in CT changes’ sources.

Limitations. Although our research achieved its purpose, the study has some weaknesses. Firstly, the sample size could influence the significance of the results. Furthermore, gender equality in the sample was not achieved, comprising more females than males in each group. As only 32 + 32 patients in both groups were included, replication studies should be done in the future, particularly with a bigger sample size and longer duration to confirm the association between phacoemulsification and choroidal thickness in eyes with and without AMD.

To sum up, not only retinal but also choroidal changes are induced following cataract surgery. Diverse factors, including inflammatory reaction, the changes in OPP, uveoscleral outflow, and activated retinal metabolism, can have an effect on CT alterations. Although the deterioration of choroidal structure and disturbed choroidal blood flow seem to play a role in AMD pathogenesis, the impact of CT change on AMD progression remains controversial in understanding the effect of cataract surgery on AMD and this change may be AL dependent [38].

## 5. Conclusions

Uneventful phacoemulsification may induce changes in the posterior eye segment. A significant increase in FRT was observed in both groups one month after cataract phacoemulsification. However, three months after the surgery, the FRT diminished without reaching the baseline value. Phacoemulsification induced CT changes in both groups. The CT increase with diverse significance in all sectors was observed one month after phacoemulsification. However, at three months postoperatively, CT results varied. The negative association between CT and age in the No-AMD group in almost all regions at all time points were disclosed. Furthermore, CT was significantly negatively associated with AL in all sectors and at all time points in the AMD group. These postoperative changes in the choroid and retina may lead not only to the late-onset PCME but also to progression of AMD. Therefore, larger studies of a longer duration should be carried out to ascertain the phacoemulsification impact on the choroidal tissues and associations with AMD.

## Figures and Tables

**Figure 1 medicina-56-00252-f001:**
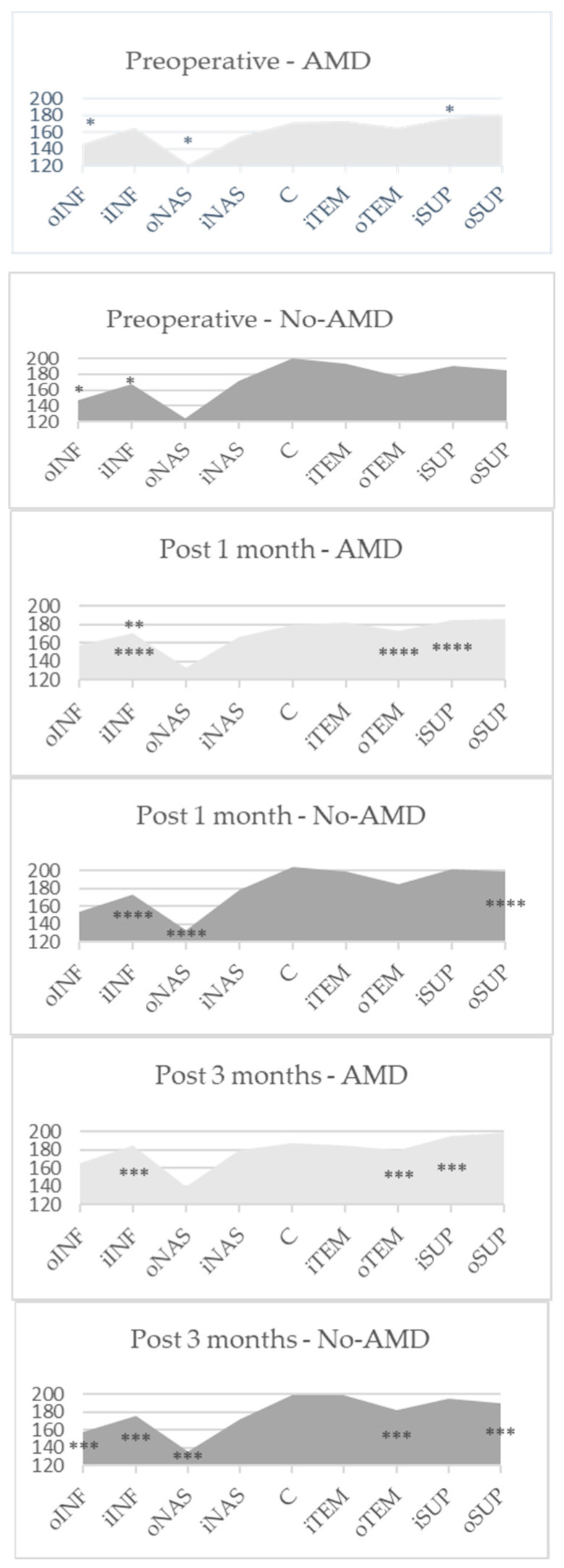
The mean automatically measured CT (μm) in EDTRS sectors in both groups. C—central ring; iNAS—nasal inner; iSUP—superior inner; iTEM—temporal inner; iINF—inferior inner; oNAS—nasal outer; oSUP—superior outer; oTEM—temporal outer; oINF—inferior outer regions. *p* *—comparison between M1 and P results, *p* **—comparison between M3 and M1 results, *p* ***—comparison between M3 and P results by Wilcoxon test. *p* ****—comparison between P, M1 and M3 results by Friedman test.

**Table 1 medicina-56-00252-t001:** Characteristics of studies analysing choroidal thickness after cataract surgery.

Author	Year	Country	N	OCT	Outcome after Cataract Surgery
**Pierru**	2014	France	115	Heidelberg	Significant ↑ in SFCT, negative correlation with age and AL
**Falcão**	2014	Portugal	14	Heidelberg	No significant changes in CT
**Bayhan**	2016	Turkey	38	RTvue-100	Significant ↑ in CT. Change in IOP correlated with the CT changes
**Yilmaz**	2016	Turkey	65	Heidelberg	Significant ↑ in SFCT
**Noda**	2014	Japan	29	Heidelberg	Significant ↑ in SFCT
**Ohsugi**	2014	Japan	100	Heidelberg	Significant ↑ in CT. Changes negatively correlated with those in IOP and AL
**Ibrahim**	2017	Egypt	53	Heidelberg	Significant ↑ in CT
**Abdellatif**	2017	Egypt	66	NIDEK	Insignificant ↑ in SFCT
**Shahzad**	2017	Pakistan	101	Topcon	Significant ↑ in SFCT
**Asena**	2017	Turkey	27	Topcon	Significant ↑ in SFCT in Phaco vs. Femto group
**Celik**	2016	Turkey	30	Zeiss	Significant ↑ in SFCT
**Jiang**	2017	China	100	Heidelberg	SFCT correlated with sex, AL, IOP, US time

SFCT—subfoveal chroidal thickness, CT—choroidal thickness, AL—axial length, IOP—intraocular pressure, US—ultrasound, N—number of eyes, OCT—optical coherence tomography, Phaco—phacoemulsification, Femto—Femto-Laser assisted cataract surgery, ↑—increase.

**Table 2 medicina-56-00252-t002:** Characteristics of study subjects.

		AMD Group	No-AMD Group	*P*
**Sex, n (%)**	**Male**	11 (34.4)	12 (37.5)	0.5 *
	**Female**	21 (65.6)	20 (62.5)	
**Age (years), M (SD)**		73.4 (7.09)	71.8 (8.96)	0.633 **
**AL (mm), M (SD)**		23.22 (1.03)	23.29 (0.94)	0.798 **
**PCME, n (%)**		1 (3.1%)	1 (3.1%)	0.754 *

AMD—age-related macular degeneration; AL—axial length; M (SD)—mean (standard deviation); PCME—pseudophakic cystoid macular edema. *—comparison of the frequencies by Fisher‘s exact test. **—comparison of the scores by Mann–Whitney U test.

**Table 3 medicina-56-00252-t003:** BCVA, IOP, OPP, FRT changes after surgery in AMD and No-AMD groups.

Group	Parameter	*P*	M1	M3	*P **	*P ***	*P ****	*P *****
**AMD**	**BCVA**	+0.44 (0.26)	+0.12 (0.33)	+0.1 (0.34)	0.001	0.391	0.001	0.001
**No-AMD**	**BCVA**	+0.35 (0.23)	+0.05 (0.13)	+0.04 (0.08)	0.001	0.440	0.001	0.001
**AMD**	**IOP**	15.87 (2.60)	14.09 (2.97)	13.27 (2.46)	0.003	0.129	0.001	0.001
**No-AMD**	**IOP**	15.16 (2.45)	13.0 (1.70)	12.8 (2.24)	0.001	0.925	0.001	0.001
**AMD**	**OPP**	56.51 (20.34)	51.63 (10.13)	52.36 (8.63)	0.233	0.182	0.789	0.231
**No-AMD**	**OPP**	65.69 (22.76)	56.13 (8.79)	57.5 (8.38)	0.07	0.586	0.484	0.531
**AMD**	**FRT**	243.17 (36.22)	262.50 (61.09)	252.36 (34.70)	0.001	0.253	0.141	0.031
**No-AMD**	**FRT**	232.63 (36.61)	246.81 (51.51)	249.10 (30.42)	0.001	0.502	0.001	0.001

AMD—group with cataract and dry age-related macular degeneration; No-AMD—group with senile cataract; BCVA—best corrected visual acuity (LogMAR); IOP—intraocular pressure (mmHg); M (SD)—mean (standard deviation); OPP—ocular perfusion pressure (mmHg); FRT—foveal retinal thickness (μm). All parameters are expressed as M (SD). *p* *—comparison between M1 and P results, *p* **—comparison between M3 and M1 results, *p* ***—comparison between M3 and P results by Wilcoxon test. *p* ****—comparison among P, M1 and M3 results by Friedman test.

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
