# Peer review of "Phacoemulsification Induced Changes of Choroidal Thickness in Eyes with Age-Related Macular Degeneration"

_medicina, 2020, doi:10.3390/medicina56050252_

Round 1
Reviewer 1 Report
In the paper yuo write: "The OCT images were obtained after pupil dilation" specify the type of mydriatic eye drops used because mydriatic drugs can change the choroidal thickness.
.
Also yuo write:" In our study, although all surgeries were uncomplicated, a PCME was observed in
3.1% patients in both groupsone month after surgery, similarly to Pierru data (2.6% patients).
I think macular edema is a complication of surgery therefore these 2 patients must be removed from the study.
The conclusions must be improved SPECIFYING HOW THIS PAPER CAN CHANGE THE APPROACH IN CLINICAL PRACTICE FOR MENAGEMENT OF ADULT CATARACT.
Reviewer 2 Report
There were few abbreviations in abstract not explained.
In Materials and Methods was not explained BCVA evaluation method.
Sample size is small, but authors notices and mentioning it.
I am lacking information and explanation of AMD characteristics - which form and level of AMD patients were diagnosed, was it same for all patients?
Reviewer 3 Report
-
Thank you for allowing me to review this study
Remarks
Title: interesting
Abstract
- My suggestion to use proffeintal English editor. ie (plenty).
- I use the help of Mrs Danielle Meyerson meyerson@gmail.com (no conflict of interest) she had done a fine job with all my works
Intro
- the first ref is “5”---this should be 1.
- please use reference manager software
- its 2020- try to find reference that are relevant to now not 10 years ago
- This is for ophthalmologist, right? I would start the intro from the “Phacoemulsification is a” = all the intro about AMD, cataract is basic- we all know it.
- if it intended to the general reader then keep it
- Choroidal structure has been evaluated in different clinical investigations with controversial results- please provide a table that summarize all these studies include author, country, year, N, definition of investigations, outcome. This will much be beneficial to the reader who could have in one table a summary of the literature.
- Subsequently, it will be easier to judge your work compared to other publications.
- Is it the first study to use Swept source-optical coherence tomograph? if so this is a novel thing. that’s why the table is important.
- what about CT in AMD? is this the first study to assess this after cataract
Methods
- why n=32 ? how this sample size was calculated?
- poor OCT quality= please define OCT quality measurement to include/exclude .
- don’t you uses NSAIDS drops post-surgery?
- χ2 may be used for larger N, I moved to Fisher exact https://www.graphpad.com/quickcalcs/contingency1/
results:
- were there and DM patients?
- AMD dry or wet?
- what about glaucoma or glaucoma tx? this is a confounder
- history of uveitis?
- what about K? IOl power?
- table 2 is not clear there are sevral p- try to think to present in a graph or table that will be readable more east
- why not use baseline, 3m, 1m etc
- add the p_value 1-4 in 4 columns not rows
- I think one graph will be great with all those numbers
- table 5--- just give summrey- no human-being can go over it
- tables- please change to graphical presentation when possible.
- try to use my methods of animation and graphics to present data:
- https://www.ncbi.nlm.nih.gov/pmc/articles/PMC5848276/
- Supplementary Animation 1 and figure 3
Discussion
- what this study adds?
- why is it important to assess the choroid? how will reading this study help promote better care? these should be discussed
- Therefore, larger studies of a longer duration should be carried out to ascertain the phacoemulsification impact on the choroidal tissues and associations with AMD- I suggest you to do this study. if not possible please add this to the limitation not at the closing argument of the paper
Reviewer 4 Report
A very interesting study, attempting to investigate the effect of phacoemulsification to choroidal thickness, implying a direct link between choroidal effusion and progression of AMD.
Some remarks:
Abstract: the conclusion section is not revealing a significant information , just duplicate the study results. A clear statement, even as a hypothesis is needed here.
Intro: the aim of the study included also other parameters as OPP and IOP changes etc
Design: there are no information about lens hardness and consumed ultrasound energy.
Has any power analysis taken place?
RESULTS: In which units VA is presented? If not LogMAR, a statistical comparison is impossible.
Table 5 should be presented as supplementary data.
Discussion: I feel that authors fail to pass a clear message regarding their results. Which is the main implication regarding findings? How can AL associated with CT and AMD progress?
The study is interesting but presentation needs to be oimproved.
Round 2
Reviewer 1 Report
The Discussion paragraph should be enhanced with clinical correlations of the Authors' results, and what these results may imply in a clinical setting.
Reviewer 3 Report
Thank you
I think the authors did a good job
Reviewer 4 Report
Authors have adequately addressed all raised issues so the study could be published after some language edit. The link between cataract surgery and AMD progression is an uncharted area and this article offers some novel information.